# Online Learning for Multivariate Hawkes Processes

Yingxiang Yang*      Jalal Etesami†      Niao He†      Negar Kiyavash*†
University of Illinois at Urbana-Champaign
Urbana, IL 61801
{yyang172,etesami2,niaohe,kiyavash} @illinois.edu

## Abstract

We develop a nonparametric and online learning algorithm that estimates the triggering functions of a multivariate Hawkes process (MHP). The approach we take approximates the triggering function $f_{i,j}(t)$ by functions in a reproducing kernel Hilbert space (RKHS), and maximizes a time-discretized version of the log-likelihood, with Tikhonov regularization. Theoretically, our algorithm achieves an $\mathcal{O}(\log T)$ regret bound. Numerical results show that our algorithm offers a competing performance to that of the nonparametric batch learning algorithm, with a run time comparable to parametric online learning algorithms.

## 1   Introduction

Multivariate Hawkes processes (MHPs) are counting processes where an arrival in one dimension can affect the arrival rates of other dimensions. They were originally proposed to statistically model the arrival patterns of earthquakes [16]. However, MHP's ability to capture mutual excitation between dimensions of a process also makes it a popular model in many other areas, including high frequency trading [3], modeling neural spike trains [24], and modeling diffusion in social networks [28], and capturing causality [12, 18].

For a $p$-dimensional MHP, the intensity function of the $i$-th dimension takes the following form:

$$\lambda_i(t) = \mu_i + \sum_{j=1}^{p} \int_0^t f_{i,j}(t-\tau)\mathrm{d}N_j(\tau), \tag{1}$$

where the constant $\mu_i$ is the base intensity of the $i$-th dimension, $N_j(t)$ counts the number of arrivals in the $j$-th dimension within $[0, t]$, and $f_{i,j}(t)$ is the triggering function that embeds the underlying causal structure of the model. In particular, one arrival in the $j$-th dimension at time $\tau$ will affect the intensity of the arrivals in the $i$-th dimension at time $t$ by the amount $f_{i,j}(t-\tau)$ for $t > \tau$. Therefore, learning the triggering function is the key to learning an MHP model. In this work, we consider the problem of estimating the $f_{i,j}(t)$s using nonparametric online learning techniques.

### 1.1   Motivations

**Why nonparametric?** Most of existing works consider exponential triggering functions:

$$f_{i,j}(t) = \alpha_{i,j} \exp\{-\beta_{i,j}t\}\mathbb{1}\{t > 0\}, \tag{2}$$

where $\alpha_{i,j}$ is unknown while $\beta_{i,j}$ is given a priori. Under this assumption, learning $f_{i,j}(t)$ is equivalent to learning a real number, $\alpha_{i,j}$. However, there are many scenarios where (2) fails to

describe the correct mutual influence pattern between dimensions. For example, [20] and [11] have reported delayed and bell-shaped triggering functions when applying the MHP model to neural spike train datasets. Moreover, when $f_{i,j}(t)$s are not exponential, or when $\beta_{i,j}$s are inaccurate, formulation in (2) is prone to model mismatch [15].

**Why online learning?** There are many reasons to consider an online framework. (i) Batch learning algorithms do not scale well due to high computational complexity [15]. (ii) The data can be costly to observe, and can be streaming in nature, for example, in criminology.

The above concerns motivate us to design an online learning algorithm in the nonparametric regime.

## 1.2 Related Works

Earlier works on learning the triggering functions can be largely categorized into three classes.

**Batch and parametric.** The simplest way to learn the triggering functions is to assume that they possess a parametric form, e.g. (2), and learn the coefficients. The most widely used estimators include the maximum likelihood estimator [23], and the minimum mean-square error estimator [2]. These estimators can also be generalized to the high dimensional case when the coefficient matrix is sparse and low-rank [2]. More generally, one can assume that $f_{i,j}(t)$s lie within the span of a given set of basis functions $\mathcal{S} = \{e_1(t), \dots, e_{|\mathcal{S}|}(t)\}$: $f_{i,j}(t) = \sum_{i=1}^{|\mathcal{S}|} c_i e_i(t)$, where $e_i(t)$s have a given parametric form [13, 27]. The state-of-the-art of such algorithms is [27], where $|\mathcal{S}|$ is adaptively chosen, which sometimes requires a significant portion of the data to determine the optimal $\mathcal{S}$.

**Batch and nonparametric.** A more sophisticated approach towards finding the set $\mathcal{S}$ is explored in [29], where the coefficients and the basis functions are iteratively updated and refined. Unlike [27], where the basis functions take a predetermined form, [29] updates the basis functions by solving a set of Euler-Lagrange equations in the nonparametric regime. However, the formulation of [29] is nonconvex, and therefore the optimality is not guaranteed. The method also requires more than $10^5$ arrivals for each dimension in order to obtain good results, on networks of less than 5 dimensions.

Another way to estimate $f_{i,j}(t)$s nonparametrically is proposed in [4], which solves a set of $p$ Wiener-Hopf systems, each of dimension at least $p^2$. The algorithm works well on small dimensions; however, it requires inverting a $p^2 \times p^2$ matrix, which is costly, if not all together infeasible, when $p$ is large.

**Online and parametric.** To the best of our knowledge, learning the triggering functions in an online setting seems largely unexplored. Under the assumption that $f_{i,j}(t)$s are exponential, [15] proposes an online algorithm using gradient descent, exploiting the evolutionary dynamics of the intensity function. The time axis is discretized into small intervals, and the updates are performed at the end of each interval. While the authors provide the online solution to the parametric case, their work cannot readily extend to the nonparametric setting where the triggering functions are not exponential, mainly because the evolutionary dynamics of the intensity functions no longer hold. Learning triggering functions nonparametrically remains an open problem.

## 1.3 Challenges and Our Contributions

Designing an online algorithm in the nonparametric regime is not without its challenges: (i) It is not clear how to represent $f_{i,j}(t)$s. In this work, we relate $f_{i,j}(t)$ to an RKHS. (ii) Although online learning with kernels is a well studied subject in other scenarios [19], a typical choice of loss function for learning an MHP usually involves the integral of $f_{i,j}(t)$s, which prevents the direct application of the representer theorem. (iii) The outputs of the algorithm at each step require a projection step to ensure positivity of the intensity function. This requires solving a quadratic programming problem, which can be computationally expensive. How to circumvent this computational complexity issue is another challenge of this work.

In this paper, we design, to the best of our knowledge, the first online learning algorithm for the triggering functions in the nonparametric regime. In particular, we tackle the challenges mentioned above, and the only assumption we make is that the triggering functions $f_{i,j}(t)$s are positive, have a decreasing tail, and that they belong to an RKHS. Theoretically, our algorithm achieves a regret

bound of $\mathcal{O}(\log T)$, and numerical experiments show that our approach outperforms the previous approaches despite the fact that they are tailored to a less general setting. In particular, our algorithm attains a similar performance to the nonparametric batch learning maximum likelihood estimators while reducing the run time extensively.

## 1.4 Notations

Prior to discussing our results, we introduce the basic notations used in the paper. Detailed notations will be introduced along the way. For a $p$-dimensional MHP, we denote the intensity function of the $i$-th dimension by $\lambda_i(t)$. We use $\boldsymbol{\lambda}(t)$ to denote the vector of intensity functions, and we use $\boldsymbol{F} = [f_{i,j}(t)]$ to denote the matrix of triggering functions. The $i$-th row of $\boldsymbol{F}$ is denoted by $\boldsymbol{f}_i$. The number of arrivals in the $i$-th dimension up to $t$ is denoted by the counting process $N_i(t)$. We set $N(t) = \sum_{i=1}^{p} N_i(t)$. The estimates of these quantities are denoted by their "hatted" versions. The arrival time of the $n$-th event in the $j$-th dimension is denoted by $\tau_{j,n}$. Lastly, define $\lfloor x \rfloor_y = y \lfloor x/y \rfloor$.

## 2 Problem Formulation

In this section, we introduce our assumptions and definitions followed by the formulation of the loss function. We omit the basics on MHPs and instead refer the readers to [22] for details.

**Assumption 2.1.** We assume that the constant base intensity $\mu_i$ is lower bounded by a given threshold $\mu_{\min} > 0$. We also assume bounded and stationary increments for the MHP [16, 9]: for any $t, z > 0$, $N_i(t) - N_i(t - z) \le \kappa_z = \mathcal{O}(z)$. See Appendix A for more details.

**Definition 2.1.** Suppose that $\{t_k\}_{k=0}^{\infty}$ is an arbitrary time sequence with $t_0 = 0$, and $\sup_{k \ge 1}(t_k - t_{k-1}) \le \delta \le 1$. Let $\varepsilon_f : [0, \infty) \to [0, \infty)$ be a continuous and bounded function such that $\lim_{t \to \infty} \varepsilon_f(t) = 0$. Then, $f(x)$ satisfies the *decreasing tail* property with *tail function* $\varepsilon_f(t)$ if

$$\sum_{k=m}^{\infty} (t_k - t_{k-1}) \sup_{x \in (t_{k-1}, t_k]} |f(x)| \le \varepsilon_f(t_{m-1}), \quad \forall m > 0.$$

**Assumption 2.2.** Let $\mathcal{H}$ be an RKHS associated with a kernel $K(\cdot, \cdot)$ that satisfies $K(x, x) \le 1$. Let $L_1[0, \infty)$ be the space of functions for which the absolute value is Lebesgue integrable. For any $i, j \in \{1, \ldots, p\}$, we assume that $f_{i,j}(t) \in \mathcal{H}$ and $f_{i,j}(t) \in L_1[0, \infty)$, with both $f_{i,j}(t)$ and $\mathrm{d}f_{i,j}(t)/\mathrm{d}t$ satisfying the decreasing tail property of Definition 2.1.

Assumption 2.1 is common and has been adopted in existing literature [22]. It ensures that the MHP is not "explosive" by assuming that $N(t)/t$ is bounded. Assumption 2.2 restricts the tail behaviors of both $f_{i,j}(t)$ and $\mathrm{d}f_{i,j}(t)/\mathrm{d}t$. Complicated as it may seem, functions with exponentially decaying tails satisfy this assumption, as is illustrated by the following example (See Appendix B for proof):

**Example 1.** Functions $f_1(t) = \exp\{-\beta t\} \mathbb{1}\{t > 0\}$ and $f_2(t) = \exp\{-(t - \gamma)^2\} \mathbb{1}\{t > 0\}$ satisfy Assumption 2.2 with tail functions $\beta^{-1} \exp\{-\beta(t - \delta)\}$ and $\sqrt{2\pi} \, \mathrm{erfc}(t/\sqrt{2} - \gamma) \exp\{\delta^2/2\}$.

### 2.1 A Discretized Loss Function for Online Learning

A common approach for learning the parameters of an MHP is to perform regularized maximum likelihood estimation. As such, we introduce a loss function comprised of the negative of the log-likelihood function and a penalty term to enforce desired structural properties, e.g. sparsity of the triggering matrix $\boldsymbol{F}$ or smoothness of the triggering functions (see, e.g., [2, 29, 27]). The negative of the log-likelihood function of an MHP over a time interval of $[0, t]$ is given by

$$\mathcal{L}_t(\boldsymbol{\lambda}) := -\sum_{i=1}^{p} \left( \int_0^t \log \lambda_i(\tau) \mathrm{d}N_i(\tau) - \int_0^t \lambda_i(\tau) \mathrm{d}\tau \right). \tag{3}$$

Let $\{\tau_1, \ldots, \tau_{N(t)}\}$ denote the arrival times of all the events within $[0, t]$ and let $\{t_0, \ldots, t_{M(t)}\}$ be a finite partition of the time interval $[0, t]$ such that $t_0 = 0$ and $t_{k+1} := \min_{\tau_i \ge t_k} \{\lfloor t_k \rfloor_\delta + \delta, \tau_i\}$. Using this partitioning, it is straightforward to see that the function in (3) can be written as

$$\mathcal{L}_t(\boldsymbol{\lambda}) = \sum_{i=1}^{p} \sum_{k=1}^{M(t)} \left( \int_{t_{k-1}}^{t_k} \lambda_i(\tau) \mathrm{d}\tau - x_{i,k} \log \lambda_i(t_k) \right) := \sum_{i=1}^{p} L_{i,t}(\lambda_i), \tag{4}$$

where $x_{i,k} := N_i(t_k) - N_i(t_{k-1})$. By the definition of $t_k$, we know that $x_{i,k} \in \{0, 1\}$. In order to learn $f_{i,j}(t)$s using an online kernel method, we require a similar result as the representer theorem in [25] that specifies the form of the optimizer. This theorem requires that the regularized version of the loss in (4) to be a function of only $f_{i,j}(t)$s. However, due to the integral part, $\mathcal{L}_t(\boldsymbol{\lambda})$ is a function of both $f_{i,j}(t)$s and their integrals, which prevents us from applying the representer theorem directly. To resolve this issue, several approaches can be applied such as adjusting the Hilbert space as proposed in [14] in context of Poisson processes, or approximating the log-likelihood function as in [15]. Here, we adopt a method similar to [15] and approximate (4) by discretizing the integral:

$$\mathcal{L}_t^{(\delta)}(\boldsymbol{\lambda}) := \sum_{i=1}^{p} \sum_{k=1}^{M(t)} ((t_k - t_{k-1})\lambda_i(t_k) - x_{i,k} \log \lambda_i(t_k)) := \sum_{i=1}^{p} L_{i,t}^{(\delta)}(\lambda_i). \tag{5}$$

Intuitively, if $\delta$ is small enough and the triggering functions are bounded, it is reasonable to expect that $L_{i,t}(\boldsymbol{\lambda})$ is close to $L_{i,t}^{(\delta)}(\boldsymbol{\lambda})$. Below, we characterize the accuracy of the above discretization and also truncation of the intensity function. First, we require the following definition.

**Definition 2.2.** We define the truncated intensity function as follows

$$\lambda_i^{(z)}(t) := \mu_i + \sum_{j=1}^{p} \int_0^t \mathbb{1}\{t - \tau < z\} f_{i,j}(t - \tau) \mathrm{d}N_j(\tau). \tag{6}$$

**Proposition 1.** Under Assumptions 2.1 and 2.2, for any $i \in \{1, \ldots, p\}$, we have

$$\left| L_{i,t}^{(\delta)}(\lambda_i^{(z)}) - L_{i,t}(\lambda_i) \right| \leq (1 + \kappa_1 \mu_{\min}^{-1}) N(t - z)\varepsilon(z) + \delta N(t)\varepsilon'(0),$$

where $\mu_{\min}$ is the lower bound for $\mu_i$, $\kappa_1$ is the upper bound for $N_i(t) - N_i(t-1)$ from Definition 2.1, while $\varepsilon$ and $\varepsilon'$ are two tail functions that uniformly capture the decreasing tail property of all $f_{i,j}(t)$s and all $\mathrm{d}f_{i,j}(t)/\mathrm{d}t$s, respectively.

The first term in the bound characterizes the approximation error when one truncates $\lambda_i(t)$ with $\lambda_i^{(z)}(t)$. The second term describes the approximation error caused by the discretization. When $z = \infty$, $\lambda_i(t) = \lambda_i^{(z)}(t)$, and the approximation error is contributed solely by discretization. Note that, in many cases, a small enough truncation error can be obtained by setting a relatively small $z$. For example, for $f_{i,j}(t) = \exp\{-3t\}\mathbb{1}\{t > 0\}$, setting $z = 10$ would result in a truncation error less than $10^{-13}$. Meanwhile, truncating $\lambda_i(t)$ greatly simplifies the procedure of computing its value. Hence, in our algorithm, we focus on $\lambda_i^{(z)}$ instead of $\lambda_i$.

In the following, we consider the regularized instantaneous loss function with the Tikhonov regularization for $f_{i,j}(t)$s and $\mu_i$:

$$l_{i,k}(\lambda_i) := (t_k - t_{k-1})\lambda_i(t_k) - x_{i,k} \log \lambda_i(t_k) + \frac{1}{2}\omega_i \mu_i^2 + \sum_{j=1}^{p} \frac{\zeta_{i,j}}{2} \|f_{i,j}\|_{\mathcal{H}}^2, \tag{7}$$

and aim at producing a sequence of estimates $\{\widehat{\lambda}_i(t_k)\}_{k=1}^{M(t)}$ of $\lambda_i(t)$ with minimal regret:

$$\sum_{k=1}^{M(t)} l_{i,k}(\widehat{\lambda}_i(t_k)) - \min_{\mu_i \geq \mu_{\min}, f_{i,j}(t) \geq 0} \sum_{k=1}^{M(t)} l_{i,k}(\lambda_i(t_k)). \tag{8}$$

Each regularized instantaneous loss function in (7) is jointly strongly convex with respect to $f_{i,j}$s and $\mu_i$. Combining with the representer theorem in [25], the minimizer to (8) is a linear combination of a finite set of kernels. In addition, by setting $\zeta_{i,j} = \mathcal{O}(1)$, our algorithm achieves $\beta$-stability with $\beta = \mathcal{O}((\zeta_{i,j}t)^{-1})$, which is typical for a learning algorithm in RKHS (Theorem 22 of [8]).

## 3 Online Learning for MHPs

We introduce our NonParametric OnLine Estimation for MHP (NPOLE-MHP) in Algorithm 1. The most important components of the algorithm are (i) the computation of the gradients and (ii) the

---

**Algorithm 1** NonParametric OnLine Estimation for MHP (NPOLE-MHP)

---

1: **input:** a sequence of step sizes $\{\eta_k\}_{k=1}^\infty$ and a set of regularization coefficients $\zeta_{i,j}$s, along with positive values of $\mu_{\min}$, $z$ and $\sigma$. **output:** $\widehat{\boldsymbol{\mu}}^{(M(t))}$ and $\widehat{\boldsymbol{F}}^{(M(t))}$.

2: Initialize $\widehat{f}_{i,j}^{(0)}$ and $\widehat{\mu}_i^{(0)}$ for all $i,j$.

3: **for** $k = 0, ..., M(t) - 1$ **do**

4:     Observe the interval $[t_k, t_{k+1})$, and compute $x_{i,k}$ for $i \in \{1, \ldots, p\}$.

5:     **for** $i = 1, \ldots, p$ **do**

6:         Set $\widehat{\mu}_i^{(k+1)} \leftarrow \max\left\{\widehat{\mu}_i^{(k)} - \eta_{k+1}\partial_{\mu_i}l_{i,k}\left(\lambda_i^{(z)}(\widehat{\mu}_i^{(k)}, \widehat{\boldsymbol{f}}_i^{(k)})\right), \mu_{\min}\right\}$.

7:         **for** $j = 1, \ldots, p$ **do**

8:             Set $\widehat{f}_{i,j}^{(k+\frac{1}{2})} \leftarrow \left[\widehat{f}_{i,j}^{(k)} - \eta_{k+1}\partial_{f_{i,j}}l_{i,k}\left(\lambda_i^{(z)}(\widehat{\mu}_i^{(k)}, \widehat{\boldsymbol{f}}_i^{(k)})\right)\right]$, and $\widehat{f}_{i,j}^{(k+1)} \leftarrow \Pi\left[\widehat{f}_{i,j}^{(k+\frac{1}{2})}\right]$.

9:         **end for**

10:     **end for**

11: **end for**

---

projections in lines 6 and 8. For the partial derivative with respect to $\mu_i$, recall the definition of $l_{i,k}$ in (7) and $\lambda_i^{(z)}$ in (6). Since $\lambda_i^{(z)}$ is a linear function of $\mu_i$, we have

$$\partial_{\mu_i}l_{i,k}\left(\lambda_i^{(z)}(\widehat{\mu}_i^{(k)}, \widehat{\boldsymbol{f}}_i^{(k)})\right) = (t_k - t_{k-1}) - x_{i,k}\left[\lambda_i^{(z)}\left(\widehat{\mu}_i^{(k)}, \widehat{\boldsymbol{f}}_i^{(k)}\right)\right]^{-1} + \omega_i\widehat{\mu}_i^{(k)} \triangleq \rho_k + \omega_i\widehat{\mu}_i^{(k)},$$

where $\rho_k$ is the simplified notation for the first two terms. Upon performing gradient descent, the algorithm makes sure that $\widehat{\mu}_i^{(k+1)} \geq \mu_{\min}$, which further ensures that $\widehat{\lambda}_i^{(z)}(\widehat{\mu}_i^{(k+1)}, \widehat{\boldsymbol{f}}_i^{(k+1)}) \geq \lambda_{\min}$.

For the update step of $\widehat{f}_{i,j}^{(k)}(t)$, notice that the $l_{i,k}$ is also a linear function with respect to $f_{i,j}$. Since $\partial_{f_{i,j}}f_{i,j}(x) = K(x, \cdot)$, which holds true due to the reproducing property of the kernel, we thus have

$$\partial_{f_{i,j}}l_{i,k}\left(\lambda_i^{(z)}(\widehat{\mu}_i^{(k)}, \widehat{\boldsymbol{f}}_i^{(k)})\right) = \rho_k \sum_{\tau_{j,n} \in [t_k - z, t_k)} K(t_k - \tau_{j,n}, \cdot) + \zeta_{i,j}\widehat{f}_{i,j}^{(k)}(\cdot). \tag{9}$$

Once again, a projection $\Pi[\cdot]$ is necessary to ensure that the estimated triggering functions are positive.

### 3.1 Projection of the Triggering Functions

For any kernel, the projection step for a triggering function can be executed by solving a quadratic programming problem: $\min \|f - \widehat{f}_{i,j}^{(k+\frac{1}{2})}\|_{\mathcal{H}}^2$ subject to $f \in \mathcal{H}$ and $f(t) \geq 0$. Ideally, the positivity constraint has to hold for every $t > 0$, but in order to simplify computation, one can approximate the solution by relaxing the constraint such that $f(t) \geq 0$ holds for only a finite set of $t$s within $[0, z]$.

**Semi-Definite Programming (SDP).** When the reproducing kernel is polynomial, the problem is much simpler. The projection step can be formulated as an SDP problem [26] as follows:

**Proposition 2.** Let $\mathcal{S} = \cup_{r \leq k}\{t_r - \tau_{j,n} : t_r - z \leq \tau_{j,n} < t_r\}$ be the set of $t_r - \tau_{j,n}$s. Let $K(x, y) = (1 + xy)^{2d}$ and $K'(x, y) = (1 + xy)^d$ be two polynomial kernels with $d \geq 1$. Furthermore, let $\mathbf{K}$ and $\mathbf{G}$ denote the Gramian matrices where the $i, j$-th element correspond to $K(s, s')$, with $s$ and $s'$ being the $i$-th and $j$-th element in $\mathcal{S}$. Suppose that $\boldsymbol{a} \in \mathbb{R}^{|S|}$ is the coefficient vector such that $\widehat{f}_{i,j}^{(k+\frac{1}{2})}(\cdot) = \sum_{s \in \mathcal{S}} a_s K(s, \cdot)$, and that the projection step returns $\widehat{f}_{i,j}^{(k+1)}(\cdot) = \sum_{s \in \mathcal{S}} b_s^* K(s, \cdot)$. Then the coefficient vector $\mathbf{b}^*$ can be obtained by

$$\mathbf{b}^* = \underset{\mathbf{b} \in \mathbb{R}^{|S|}}{\operatorname{argmin}} -2\boldsymbol{a}^\top \mathbf{K}\mathbf{b} + \mathbf{b}^\top \mathbf{K}\mathbf{b}, \quad \text{s.t.} \ \mathbf{G} \cdot \operatorname{diag}(\mathbf{b}) + \operatorname{diag}(\mathbf{b}) \cdot \mathbf{G} \succeq 0. \tag{10}$$

**Non-convex approach.** Alternatively, we can assume that $f_{i,j}(t) = g_{i,j}^2(t)$ where $g_{i,j}(t) \in \mathcal{H}$. By minimizing the loss with respect to $g_{i,j}(t)$, one can naturally guarantee that $f_{i,j}(t) \geq 0$. This method was adopted in [14] for estimating the intensity function of non-homogeneous Poisson processes. While this approach breaks the convexity of the loss function, it works relatively well when the initialization is close to the global minimum. It is also interestingly related to a line of recent works in non-convex SDP [6], as well as phase retrieval with Wirtinger flow [10]. Deriving guarantees on regret bound and convergence performances is a future direction implied by the result of this work.

## 4 Theoretical Properties

We now discuss the theoretical properties of NPOLE-MHP. We start with defining the regret.

**Definition 4.1.** The regret of Algorithm 1 at time $t$ is given by

$$R_t^{(\delta)}(\lambda_i^{(z)}(\mu_i, \boldsymbol{f}_i)) := \sum_{k=1}^{M(t)} \left( l_{i,k}(\lambda_i^{(z)}(\widehat{\mu}_i^{(k)}, \widehat{\boldsymbol{f}}_i^{(k)})) - l_{i,k}(\lambda_i^{(z)}(\mu_i, \boldsymbol{f}_i)) \right),$$

where $\widehat{\mu}_i^{(k)}$ and $\widehat{\boldsymbol{f}}_i^{(k)}$ denote the estimated base intensity and the triggering functions, respectively.

**Theorem 1.** Suppose that the observations are generated from a $p$-dimensional MHP that satisfies Assumptions 2.1 and 2.2. Let $\zeta = \min_{i,j}\{\zeta_{i,j}, \omega_i\}$, and $\eta_k = 1/(\zeta k + b)$ for some positive constants $b$. Then

$$R_t^{(\delta)}(\lambda_i^{(z)}(\mu_i, \boldsymbol{f}_i)) \leq C_1(1 + \log M(t)),$$

where $C_1 = 2(1 + p\kappa_z^2)\zeta^{-1}|\delta - \mu_{\min}^{-1}|^2$.

The regret bound of Theorem 1 resembles the regret bound for a typical online learning algorithm with strongly convex loss function (see for example, Theorem 3.3 of [17]). When $\delta$, $\zeta$ and $\mu_{\min}^{-1}$ are fixed, $C_1 = \mathcal{O}(p)$, which is intuitive as one needs to update $p$ functions at a time. Note that the regret in Definition 4.1, encodes the performance of Algorithm 1 by comparing its loss with the approximated loss. Below, we compare the loss of Algorithm 1 with the original loss in (4).

**Corollary 1.** Under the same assumptions as Theorem 1, we have

$$\sum_{k=1}^{M(t)} \left( l_{i,k}(\lambda_i^{(z)}(\widehat{\mu}_i, \widehat{\boldsymbol{f}}_i^{(k)})) - l_{i,k}(\lambda_i(\mu_i, \boldsymbol{f}_i)) \right) \leq C_1[1 + \log M(t)] + C_2 N(t), \tag{11}$$

where $C_1$ is defined in Theorem 1 and $C_2 = (1 + \kappa_1\mu_{\min}^{-1})\varepsilon(z) + \delta\varepsilon'(0)$.

Note that $C_3 N(t)$ is due to discretization and truncation steps and it can be made arbitrary small for given $t$ and setting small $\delta$ and large enough $z$.

**Computational Complexity.** Since $\widehat{f}_i$s can be estimated in parallel, we restrict our analysis to the case of a fixed $i \in \{1, \ldots, p\}$ in a single iteration. For each iteration, the computational complexity comes from evaluating the intensity function and projection. Since the number of arrivals within the interval $[t_k - z, t_k)$ is bounded by $p\kappa_z$ and $\kappa_z = \mathcal{O}(1)$, evaluating the intensity costs $\mathcal{O}(p^2)$ operations. For the projection in each step, one can truncate the number of kernels used to represent $f_{i,j}(t)$ to be $\mathcal{O}(1)$ with controllable error (Proposition 1 of [19]), and therefore the computation cost is $\mathcal{O}(1)$. Hence, the per iteration computation cost of NPOLE-MHP is $\mathcal{O}(p^2)$. By comparison, parametric online algorithms (DMD, OGD of [15]) also require $\mathcal{O}(p^2)$ operations for each iteration, while the batch learning algorithms (MLE-SGLP, MLE of [27]) require $\mathcal{O}(p^2 t^3)$ operations.

## 5 Numerical Experiments

We evaluate the performance of NPOLE-MHP on both synthetic and real data, from multiple aspects: (i) visual assessment of the goodness-of-fit comparing to the ground truth; (ii) the "average $L_1$ error" defined as the average of $\sum_{i=1}^{p}\sum_{j=1}^{p}\|f_{i,j} - \widehat{f}_{i,j}\|_{L_1[0,z]}$ over multiple trials; (iii) scalability over both dimension $p$ and time horizon $T$. For benchmarks, we compare NPOLE-MHP's performance to that of online parametric algorithms (DMD, OGD of [15]) and nonparametric batch learning algorithms (MLE-SGLP, MLE of [27]).

### 5.1 Synthetic Data

Consider a 5-dimensional MHP with $\mu_i = 0.05$ for all dimensions. We set the triggering functions as

$$\boldsymbol{F} = \begin{bmatrix} e^{-2.5t} & 0 & 0 & e^{-10(t-1)^2} & 0 \\ 2^{-5t} & (1+\cos(\pi t))e^{-t}/2 & e^{-5t} & 0 & 0 \\ 0 & 2e^{-3t} & 0 & 0 & 0 \\ 0 & 0 & 0 & 0.6e^{-3t^2} + 0.4e^{-3(t-1)^2} & e^{-4t} \\ 0 & 0 & te^{-5(t-1)^2} & 0 & e^{-3t} \end{bmatrix}.$$

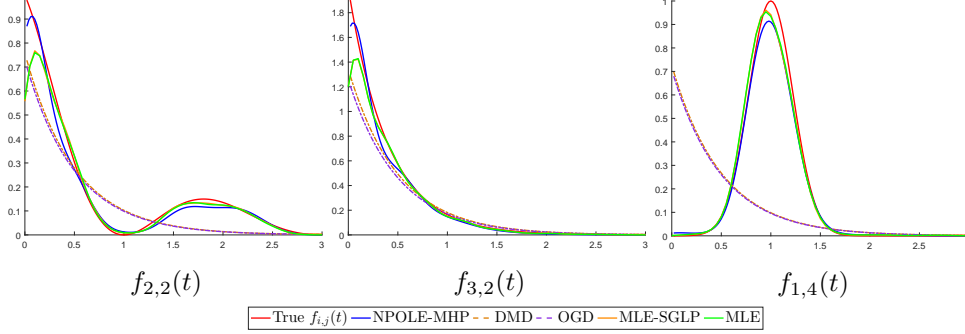

Figure 1: Performances of different algorithms for estimating $\boldsymbol{F}$. Complete set of result can be found in Appendix F. For each subplot, the horizontal axis covers $[0, z]$ and the vertical axis covers $[0, 1]$. The performances are similar between DMD and OGD, and between MLE and MLE-SGLP.

The design of $\boldsymbol{F}$ allows us to test NPOLE-MHP's ability of detecting (i) exponential triggering functions with various decaying rate; (ii) zero functions; (iii) functions with delayed peaks and tail behaviors different from an exponential function.

**Goodness-of-fit.** We run NPOLE-MHP over a set of data with $T = 10^5$ and around $4 \times 10^4$ events for each dimension. The parameters are chosen by grid search over a small portion of data, and the parameters of the benchmark algorithms are fine-tuned (see Appendix F for details). In particular, we set the discretization level $\delta = 0.05$, the window size $z = 3$, the step size $\eta_k = (k\delta/20 + 100)^{-1}$, and the regularization coefficient $\zeta_{i,j} \equiv \zeta = 10^{-8}$. The performances of NPOLE-MHP and benchmarks are shown in Figure 1. We see that NPOLE-MHP captures the shape of the function much better than the DMD and OGD algorithms with mismatched forms of the triggering functions. It is especially visible for $f_{1,4}(t)$ and $f_{2,2}(t)$. In fact, our algorithm scores a similar performance to the batch learning MLE estimator, which is optimal for any given set of data. We next plot the average loss per iteration for this dataset in Figure 2. In the left-hand side, the loss is high due to initialization. However, the effect of initialization quickly diminishes as the number of events increases.

**Run time comparison.** The simulation of the DMD and OGD algorithms took 2 minutes combined on a Macintosh with two 6-core Intel Xeon processor at 2.4 GHz, while NPOLE-MHP took 3 minutes. The batch learning algorithms MLE-SGLP and MLE in [27] each took about 1.5 hours. Therefore, our algorithm achieves the performance similar to batch learning algorithms with a run time close to that of parametric online learning algorithms.

**Effects of the hyperparameters:** $\delta$, $\zeta_{i,j}$, **and** $\eta_k$**.** We investigate the sensitivity of NPOLE-MHP with respect to the hyperparameters, measuring the "averaged $L_1$ error" defined at the beginning of this section. We independently generate 100 sets of data with the same parameters, and a smaller $T = 10^4$ for faster data generation. The result is shown in Table 1. For NPOLE-MHP, we fix $\eta_k = 1/(k/2000 + 10)$. MLE and MLE-SGLP score around 1.949 with 5/5 inner/outer rounds of iterations. NPOLE-MHP's performance is robust when the regularization coefficient and discretization level are sufficiently small. It surpasses MLE and MLE-SGLP on large datasets, in which case the iterations of MLE and MLE-SGLP are limited due to computational considerations. As $\zeta$ increases, the error decreases first before rising drastically, a phenomenon caused by the mismatch between the loss functions. For the step size, the error varies under different choice of $\eta_k$, which can be selected via grid-search on a small portion of the data like most other online algorithms.

### 5.2 Real Data: Inferencing Impact Between News Agencies with Memetracker Data

We test the performance of NPOLE-MHP on the memetracker data [21], which collects from the internet a set of popular phrases, including their content, the time they were posted, and the url address of the articles that included them. We study the relationship between different news agencies, modeling the data with a $p$-dimensional MHP where each dimension corresponds to a news website. Unlike [15], which conducted a similar experiment where all the data was used, we focus on only 20

|   |      | Regularization $\log_{10} \zeta$ | | | | |
|---|------|------|------|------|------|------|
|   |      | $-8$ | $-6$ | $-4$ | $-2$ | $0$ |
|   | 0.01 | 1.83 | 1.83 | 1.84 | 4.15 | 4.64 |
|   | 0.05 | 1.86 | 1.86 | 1.86 | 3.10 | 4.64 |
| $\delta$ | 0.1 | 1.92 | 1.92 | 1.88 | 2.73 | 4.64 |
|   | 0.5 | 4.80 | 4.80 | 4.64 | 2.19 | 4.62 |
|   | 1    | 5.73 | 5.73 | 5.58 | 2.38 | 4.59 |

Table 1: Effect of hyperparameters $\zeta$ and $\delta$, measured by the "average $L_1$ error".

|   |     | Horizon $T$ (days) | | |
|---|-----|------|------|------|
|   |     | 1.8  | 3.6  | 5.4  |
|   | 20  | 3.9  | 9.1  | 15.3 |
|   | 40  | 4.6  | 10.4 | 17.0 |
| Dimension $p$ | 60  | 4.6  | 10.2 | 16.7 |
|   | 80  | 4.5  | 10.0 | 16.4 |
|   | 100 | 4.5  | 9.7  | 15.9 |

Table 2: Average CPU-time for estimating one triggering function (seconds).

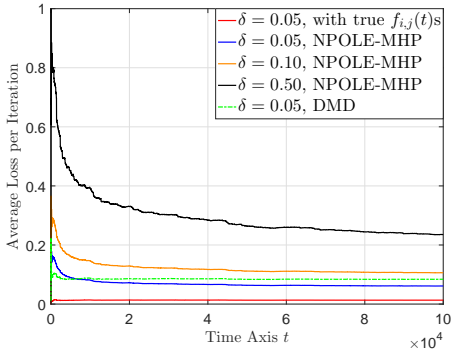

Figure 2: Effect of discretization in NPOLE-MHP.

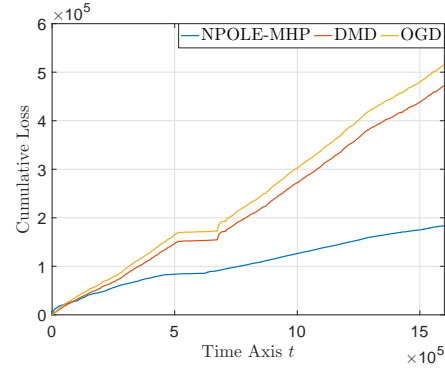

Figure 3: Cumulative loss on memetracker data of 20 dimensions.

websites that are most active, using 18 days of data. We plot the cumulative losses in Figure 3, using a window size of 3 hours, an update interval $\delta = 0.2$ seconds, and a step size $\eta_k = 1/(k\zeta + 800)$ with $\zeta = 10^{-10}$ for NPOLE-MHP. For DMD and OGD, we set $\eta_k = 5/\sqrt{T/\delta}$. The result shows that NPOLE-MHP accumulates a smaller loss per step compared to OGD and DMD.

**Scalability and generalization error.** Finally, we evaluate the scalability of NPOLE-MHP using the average CPU-time for estimating one triggering function. The result in Table 2 shows that the computation cost of NPOLE-MHP scales almost linearly with the dimension and data size. When scaling the data to 100 dimensions and $2 \times 10^5$ events, NPOLE-MHP scores an average 0.01 loss per iteration on both training and test data, while OGD and DMD scored 0.005 on training data and 0.14 on test data. This shows a much better generalization performance of NPOLE-MHP.

# 6   Conclusion

We developed a nonparametric method for learning the triggering functions of a multivariate Hawkes process (MHP) given time series observations. To formulate the instantaneous loss function, we adopted the method of discretizing the time axis into small intervals of lengths at most $\delta$, and we derived the corresponding upper bound for approximation error. From this point, we proposed an online learning algorithm, NPOLE-MHP, based on the framework of online kernel learning and exploits the interarrival time statistics under the MHP setup. Theoretically, we derived the regret bound for NPOLE-MHP, which is $\mathcal{O}(\log T)$ when the time horizon $T$ is known a priori, and we showed that the per iteration cost of NPOLE-MHP is $\mathcal{O}(p^2)$. Numerically, we compared NPOLE-MHP's performance with parametric online learning algorithms and nonparametric batch learning algorithms. Results on both synthetic and real data showed that we are able to achieve similar performance to that of the nonparametric batch learning algorithms with a run time comparable to the parametric online learning algorithms.

## Footnotes

*Department of Electrical and Computer Engineering. †Department of Industrial and Enterprise Systems Engineering. This work was supported in part by MURI grant ARMY W911NF-15-1-0479 and ONR grant W911NF-15-1-0479.

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
