[Supplementary Material · supplementary.pdf]

# A   Boundedness of the Number of Arrivals

For an MHP with stationary increments, we have $\mathbb{E}[\mathrm{d}N_i(t)|\mathcal{F}^t] = \lambda_i(t)\mathrm{d}t$, where $\mathcal{F}^t$ denotes the $\sigma$-algebra generated by $\{N_1(t), ..., N_p(t)\}$. This implies

$$\mathbb{E}[N_i(t) - N_i(t - z)] = \bar{\lambda}_i z,$$

where $\bar{\lambda}_i := \mathbb{E}[\lambda_i(t)]$ is a positive constant. On the other hand, using the second order statistics of Hawkes processes in [2], we know that the covariance matrix of Hawkes process is given by

$$\mathbb{E}\left[(\mathbf{N}(z) - \mathbf{N}(0) - \bar{\boldsymbol{\lambda}}z)(\mathbf{N}(t) - \mathbf{N}(t - z) - \bar{\boldsymbol{\lambda}}z)^\top\right] = \Psi_t z,$$

where $\Psi_t$ is a $p \times p$ matrix given in Theorem 1 of [1], $\mathbf{N}(z) = [N_1(z), \ldots, N_p(z)]^\top$ and $\bar{\boldsymbol{\lambda}} = [\lambda_1, \ldots, \lambda_p]^\top$. Hence, we obtain that, with high probability, $N_i(t) - N_i(t - z) = \Theta(z)$.

# B   Proof of Example 1

Notice that $f_{i,j}^{(0)}(t)$ is a monotonically decreasing function. Therefore, for any $\delta$-update set, we have

$$\sup_{x \in (t_{k-1}, t_k]} |f_{i,j}^{(0)}(x)| = f_{i,j}^{(0)}(t_{k-1}) = \exp\{-\beta t_{k-1}\}.$$

Hence,

$$
\begin{aligned}
(t_k - t_{k-1}) \exp\{-\beta t_{k-1}\} &\leq \int_{t_{k-1}}^{t_k} \exp\{-\beta(t - \delta)\}\mathrm{d}t \\
&= \frac{1}{\beta}\left(\exp\{\beta(\delta - t_{k-1})\} - \exp\{\beta(\delta - t_k)\}\right),
\end{aligned}
$$

where the inequality is due to the fact that for $t_k \leq t_{k-1} + \delta$ and $t \in [t_{k-1}, t_k]$, $\exp\{-\beta t_{k-1}\} \leq \exp\{-\beta(t - \delta)\}$. Summing up both sides of the above inequality implies

$$\sum_{k=m}^{\infty}(t_k - t_{k-1})\sup_{x \in (t_{k-1}, t_k]}|f_{i,j}^{(0)}(x)| \leq \frac{1}{\beta}\exp\{-\beta(t_{m-1} - \delta)\}.$$

Similarly, one can obtain the tail functions of $|\mathrm{d}f_{i,j}^{(0)}(t)/\mathrm{d}t|$ that is $\exp\{-\beta(t_{m-1} - \delta)\}$. In case of $f_{i,j}^{(1)}(t)$, for $t_{m-1} > \gamma + 2\delta$, we have

$$(t_k - t_{k-1})\sup_{t \in [t_{k-1}, t_k]}e^{-(t-\gamma)^2} \leq \int_{t_{k-1}}^{t_k}e^{-\frac{1}{2}(t-\gamma)^2 + \frac{\delta^2}{2}}\mathrm{d}t,$$

in which we used the fact that $t_k \leq t_{k-1} + \delta$. Hence, a tail function for $f_{i,j}^{(1)}(t)$ is

$$\int_{t_{m-1}}^{\infty}e^{-\frac{1}{2}(t-\gamma)^2 + \frac{\delta^2}{2}}\mathrm{d}t = \sqrt{\frac{\pi}{2}}\mathrm{erfc}\left(\frac{t_{m-1}}{\sqrt{2}} - \gamma\right)e^{\frac{\delta^2}{2}}.$$

For the derivative of $f_{i,j}^{(1)}(t)$ and for $t_{m-1} > \gamma + 2\delta + 1/\sqrt{2}$, we have

$$(t_k - t_{k-1})\sup_{t \in [t_{k-1}, t_k]}|t - \gamma|e^{-(t-\gamma)^2} \leq \int_{t_{k-1}}^{t_k}(t - \gamma)e^{-\frac{1}{2}(t-\gamma)^2 + \frac{\delta^2}{2}}\mathrm{d}t.$$

This implies the following tail function:

$$\varepsilon(t) = e^{\frac{\delta^2}{2}}\int_{t_{m-1}}^{\infty}(t - \gamma)e^{-\frac{1}{2}(t-\gamma)^2}\mathrm{d}t.$$

## C   Proof of Proposition 1

Fix the triggering functions $\boldsymbol{f}_i$ and the constant base intensity $\mu_i$. Then,

$$\left| L_{i,t}^{(\delta)}(\lambda_i^{(z)}) - L_{i,t}(\lambda_i) \right| \le \left| L_{i,t}^{(\delta)}(\lambda_i^{(z)}) - L_{i,t}^{(\delta)}(\lambda_i) \right| + \left| L_{i,t}^{(\delta)}(\lambda_i) - L_{i,t}(\lambda_i) \right|. \tag{12}$$

We bound the first term on the right-hand side that is corresponding to the truncation error as follows:

$$\left| L_{i,t}^{(\delta)}(\lambda_i^{(z)}) - L_{i,t}^{(\delta)}(\lambda_i) \right| = \left| \sum_{k=1}^{M(t)} (t_k - t_{k-1}) \left( \lambda_i^{(z)}(t_k) - \lambda_i(t_k) \right) - \sum_{k=1}^{M(t)} x_{i,k} \left( \log \lambda_i^{(z)}(t_k) - \log \lambda_i(t_k) \right) \right|$$

$$\le \left| \sum_{k=1}^{M(t)} (t_k - t_{k-1}) \left( \lambda_i^{(z)}(t_k) - \lambda_i(t_k) \right) \right| + \left| \sum_{k=1}^{M(t)} \frac{x_{i,k}}{\mu_{\min}} \left( \lambda_i^{(z)}(t_k) - \lambda_i(t_k) \right) \right|$$

$$\le \sum_{k=1}^{M(t)} (t_k - t_{k-1} + \frac{x_{i,k}}{\mu_{\min}}) \left| \lambda_i^{(z)}(t_k) - \lambda_i(t_k) \right|. \tag{13}$$

First, we define $\alpha(t) := \mathbb{1}\{t \le z\}$. Using the fact that $1 - \alpha(t_k - \tau_{j,n}) = \mathbb{1}\{t_k - \tau_{j,n} > z\}$, we obtain

$$\sum_{k=1}^{M(t)} (t_k - t_{k-1}) \left| \lambda_i^{(z)}(t_k) - \lambda_i(t_k) \right| = \sum_{k=1}^{M(t)} \sum_{j=1}^{p} \sum_{n=1}^{N_j(t_k)} (t_k - t_{k-1}) f_{i,j}(t_k - \tau_{j,n})(1 - \alpha(t_k - \tau_{j,n}))$$

$$= \sum_{j=1}^{p} \sum_{k:t_k \in [z,t)} \sum_{\tau_{j,n} \in [0,t_k - z)} (t_k - t_{k-1}) f_{i,j}(t_k - \tau_{j,n})$$

$$= \sum_{j=1}^{p} \sum_{\tau_{j,n} \in [0,t-z)} \sum_{k:t_k - \tau_{j,n} \ge z} (t_k - \tau_{j,n} - t_{k-1} + \tau_{j,n}) f_{i,j}(t_k - \tau_{j,n})$$

$$\le \sum_{j=1}^{p} \sum_{\tau_{j,n} \in [0,t-z)} \varepsilon_{f_{i,j},\delta}(z) \le \sum_{j=1}^{p} N_j(t-z)\varepsilon(z)$$

$$= N(t-z)\varepsilon(z), \tag{14}$$

where $\varepsilon(t)$ is a tail function such that for any $i$ and $j$, $\varepsilon_{f_{i,j},\delta}(t) \le \varepsilon(t)$. The above inequality is due to Assumption 2.2. Suppose that the $m$-th arrival of the $i$-th dimension is in $[t_{k_m - 1}, t_{k_m})$. Then,

$$\sum_{k=1}^{M(t)} x_{i,k} \left| \lambda_i^{(z)}(t_k) - \lambda_i(t_k) \right| = \sum_{m=1}^{N_i(t)} \left| \lambda_i^{(z)}(t_{k_m}) - \lambda_i(t_{k_m}) \right|$$

$$= \sum_{m=1}^{N_i(t)} \sum_{j=1}^{p} \sum_{n=1}^{N_j(t_{k_m})} f_{i,j}(t_{k_m} - \tau_{j,n})(1 - \alpha(t_{k_m} - \tau_{j,n}))$$

$$= \sum_{j=1}^{p} \sum_{m:z < t_{k_m} < t} \sum_{\tau_{j,n} \in [0,t_{k_m} - z)} f_{i,j}(t_{k_m} - \tau_{j,n})$$

$$\le \sum_{j=1}^{p} \sum_{\tau_{j,n} \in [0,t-z)} \sum_{m:t_{k_m} - \tau_{j,n} > z} f_{i,j}(t_{k_m} - \tau_{j,n})$$

$$\le \sum_{j=1}^{p} \sum_{\tau_{j,n} \in [0,t-z)} \kappa_1 \varepsilon(z) = N(t-z)\kappa_1 \varepsilon(z). \tag{15}$$

The last inequality uses Assumption 2.2 and the fact that the number of arrivals in an interval of length one is bounded by $\kappa_1$. Therefore, by combining (13) with (14) and (15), we get

$$\left| L_{i,t}^{(\delta)}(\lambda_i^{(z)}) - L_{i,t}^{(\delta)}(\lambda_i) \right| \le \left( 1 + \frac{\kappa_1}{\mu_{\min}} \right) N(t-z)\varepsilon(z). \tag{16}$$

We now proceed to bound the second term in (12). By the definition, we have

$$\left| L_{i,t}^{(\delta)}(\lambda_i) - L_{i,t}(\lambda_i) \right| = \left| \sum_{k=1}^{M(t)} (t_k - t_{k-1})\lambda_i(t_k) - \int_0^t \lambda_i(\tau)\mathrm{d}\tau \right|. \tag{17}$$

To bound the right-hand side, using the definition of $\lambda_i$, we have that (17) is bounded above by

$$\sum_{k=1}^{M(t)} \sum_{j=1}^{p} \left[ \left| \int_{t_{k-1}}^{t_k} \sum_{\tau_{j,n}<\tau} f_{i,j}(\tau - \tau_{j,n}) - \sum_{\tau_{j,n}<t_k} f_{i,j}(t_k - \tau_{j,n})\mathrm{d}\tau \right| \right]$$

$$\overset{(a)}{=} \sum_{k=1}^{M(t)} \sum_{j=1}^{p} \left[ \left| \int_{t_{k-1}}^{t_k} \sum_{\tau_{j,n}<t_k} [f_{i,j}(\tau - \tau_{j,n}) - f_{i,j}(t_k - \tau_{j,n})]\mathrm{d}\tau \right| \right]$$

$$\leq \sum_{j=1}^{p} \sum_{k=1}^{M(t)} \left[ \sum_{\tau_{j,n}<t_k} (t_k - t_{k-1})^2 \sup_{x\in(t_{k-1}-\tau_{j,n},t_k-\tau_{j,n}]} \left| \frac{\mathrm{d}f_{i,j}(x)}{\mathrm{d}x} \right| \right]$$

$$= \sum_{j=1}^{p} \sum_{\tau_{j,n}<t} \sum_{t_k\geq\tau_{j,n}} (t_k - t_{k-1})^2 \sup_{x\in(t_{k-1}-\tau_{j,n},t_k-\tau_{j,n}]} \left| \frac{\mathrm{d}f_{i,j}(x)}{\mathrm{d}x} \right|$$

$$\leq \sum_{j=1}^{p} \sum_{\tau_{j,n}<t} \delta\varepsilon_{f'_{i,j},\delta}(\tau_{j,n}) \overset{(b)}{\leq} \delta N(t)\varepsilon'(0),$$

where $\varepsilon'(t)$ is a tail function such that for any $i$ and $j$, $\varepsilon_{f'_{i,j},\delta}(t) \leq \varepsilon'(t)$. In the above equations, (a) uses the fact that in an interval $[t_{k-1}, t_k]$, arrivals can only happen at the endpoints. Moreover, (b) uses Assumption 2.2. Using the upper bounds of (17) and (16) in (12) will imply the result.

## D  Proof of Theorem 1

We prove this regret bound following the proof technique for Theorem 4 of [19] and the proof technique for Theorem 3.3 of [17]. The outline of this proof is as follows:

- Firstly, we derive the following upper bound:

$$\sum_{k=1}^{M(t)} \left( l_{i,k}[\lambda_i^{(z)}(\widehat{\mu}_i^{(k)}, \widehat{\boldsymbol{f}}_i^{(k)})] - l_{i,k}[\lambda_i^{(z)}(\mu_i, \widehat{\boldsymbol{f}}_i^{(k)})] \right) \leq 2\zeta^{-1}|\delta - \mu_{\min}^{-1}|^2(1 + \log M(t)). \tag{18}$$

- Next, we derive the following upper bound:

$$\sum_{k=1}^{M(t)} \left( l_{i,k}[\lambda_i^{(z)}(\mu_i, \widehat{\boldsymbol{f}}_i^{(k)})] - l_{i,k}[\lambda_i^{(z)}(\mu_i, \boldsymbol{f}_i)] \right) \leq 2p\kappa_z^2\zeta^{-1}|\delta - \mu_{\min}^{-1}|^2(1 + \log M(t)). \tag{19}$$

  To do this, we need three separate steps:

  - Prove the Lemma 1, which we state below.
  - Prove that the instantaneous loss function is strongly convex with respect to $\widehat{\boldsymbol{f}}_i^{(k)}$ and $\|\cdot\|_{\mathcal{H}}^2$.
  - Use the result of Lemma 1 and apply the proof procedure of Theorem 4 of [19] and Theorem 3.3 of [17].

- Lastly, we combine the results of (18) and (19) to obtain the regret bound:

$$R_t^{(\delta)}[\lambda_i^{(z)}(\mu_i, \boldsymbol{f}_i)) \leq C_1(1 + \log M(t)], \tag{20}$$

  where $C_1 = 2(1 + p\kappa_z^2)|\delta - \mu_{\min}^{-1}|^2$.

**Step 0: Technical assumptions and lemma.** Before the main body of the proof, we need to introduce the following technical assumption, as well as a lemma that bounds the $\mathcal{H}$-norm of $\partial_{f_{i,j}} l_{i,k}$. These result will be frequently referred to throughout the main body of the proof.

**Assumption D.1.** We assume that $\delta$ is set small enough such that $|\delta - \mu_{\min}^{-1}| > \delta$.

This assumption does not affect the implementation of the algorithm since $\delta$ and $\mu_{\min}$ are both manually set.

**Assumption D.2.** We assume that the initialization of the algorithm is nice enough:

$$\widehat{\mu}_i^{(0)} \leq \omega_i^{-1} |\delta - \mu_{\min}^{-1}|,$$

and

$$\left\| \widehat{f}_{i,j}^{(0)} \right\|_{\mathcal{H}} \leq \kappa_z \zeta_{i,j}^{-1} \left| \delta - \mu_{\min}^{-1} \right|.$$

Similar to D.1, this assumption does not affect the implementation of the algorithm as we can set $\mu_{\min}$ to be small.

The following lemma is needed in Step 2, and will be proved in Step 2.

**Lemma 1.** Suppose that Assumptions 2.1 and 2.2 hold. Then, for any $i, j, k$, the intermediate output of Algorithm 1 at step $k$ satisfies

$$\left\| \partial_{f_{i,j}} l_{i,k} \left( \lambda_i^{(z)}(\widehat{\mu}_i^{(k)}, \widehat{f}_i^{(k)}) \right) \right\|_{\mathcal{H}} \leq \left\{ \begin{array}{ll} 2 \left| \delta - \mu_{\min}^{-1} \right| \kappa_z & \text{if } x_{i,k} = 1 \\ 2\delta \kappa_z & \text{if } x_{i,k} = 0 \end{array} \right.,$$

and

$$\left\| \widehat{f}_{i,j}^{(k)} \right\|_{\mathcal{H}} \leq \zeta_{i,j}^{-1} \kappa_z \left| \delta - \mu_{\min}^{-1} \right|.$$

We are now ready to prove the main part of the theorem.

**Step 1: Proving equation** (18). We start the proof of (18) by observing the following fact: given $\widehat{f}_i^{(k)}$, the loss function $l_{i,k}(\lambda_i^{(z)}(\widehat{\mu}_i^{(k)}, \widehat{f}_i^{(k)}))$ is $\omega_i$-strongly convex with respect to $\widehat{\mu}_i$ and the square operator. This implies that

$$l_{i,k}[\lambda_i^{(z)}(\mu_i, \widehat{f}_i^{(k)})] \geq l_{i,k}[\lambda_i^{(z)}(\widehat{\mu}_i^{(k)}, \widehat{f}_i^{(k)})] + \left\langle \partial_{\mu_i} l_{i,k}[\lambda_i^{(z)}(\widehat{\mu}_i^{(k)}, \widehat{f}_i^{(k)})], \mu_i - \widehat{\mu}_i^{(k)} \right\rangle +$$
$$+ \frac{\omega_i}{2}(\mu_i - \widehat{\mu}_i^{(k)})^2,$$

which further indicates that

$$2l_{i,k}[\lambda_i^{(z)}(\widehat{\mu}_i^{(k)}, \widehat{f}_i^{(k)})] - 2l_{i,k}[\lambda_i^{(z)}(\mu_i, \widehat{f}_i^{(k)})] \leq 2 \left\langle \partial_{\mu_i} l_{i,k}[\lambda_i^{(z)}(\widehat{\mu}_i^{(k)}, \widehat{f}_i^{(k)})], \widehat{\mu}_i^{(k)} - \mu_i \right\rangle -$$
$$- \omega_i(\mu_i - \widehat{\mu}_i^{(k)})^2. \tag{21}$$

By the update rule, we have

$$\widehat{\mu}_i^{(k+1)} = \Pi \left[ \widehat{\mu}_i^{(k)} - \eta_k \partial_{\mu_i} l_{i,k}[\lambda_i^{(z)}(\widehat{\mu}_i^{(k)}, \widehat{f}_i^{(k)})] \right].$$

Since the projection is contractive, we have

$$\begin{aligned} \left( \widehat{\mu}_i^{(k+1)} - \mu_i \right)^2 &\leq \left( \widehat{\mu}_i^{(k+\frac{1}{2})} - \mu_i \right)^2 \\ &= \left( \widehat{\mu}_i^{(k)} - \eta_k \partial_{\mu_i} l_{i,k}[\lambda_i^{(z)}(\widehat{\mu}_i^{(k)}, \widehat{f}_i^{(k)})] - \mu_i \right)^2 \\ &= \left( \widehat{\mu}_i^{(k)} - \mu_i \right)^2 - 2\eta_k \left\langle \widehat{\mu}_i^{(k)} - \mu_i, l_{i,k}[\lambda_i^{(z)}(\widehat{\mu}_i^{(k)}, \widehat{f}_i^{(k)})] \right\rangle + \\ &+ \eta_k^2 \left[ \partial_{\mu_i} l_{i,k}[\lambda_i^{(z)}(\widehat{\mu}_i^{(k)}, \widehat{f}_i^{(k)})] \right]^2. \end{aligned}$$

Hence,

$$2\left\langle \widehat{\mu}_i^{(k)} - \mu_i, l_{i,k}[\lambda_i^{(z)}(\widehat{\mu}_i^{(k)}, \widehat{\boldsymbol{f}}_i^{(k)})]\right\rangle \leq \frac{1}{\eta_k}\left[\left(\widehat{\mu}_i^{(k)} - \mu_i\right)^2 - \left(\widehat{\mu}_i^{(k+1)} - \mu_i\right)^2\right] +$$
$$+ \eta_k\left[\partial_{\mu_i} l_{i,k}[\lambda_i^{(z)}(\widehat{\mu}_i^{(k)}, \widehat{\boldsymbol{f}}_i^{(k)})]\right]^2. \tag{22}$$

It is not hard to verify that $|\partial_{\mu_i} l_{i,k}[\lambda_i^{(z)}(\widehat{\mu}_i^{(k)}, \widehat{\boldsymbol{f}}_i^{(k)})]|$ is bounded: first, notice that

$$\left|\partial_{\mu_i} l_{i,k}[\lambda_i^{(z)}(\widehat{\mu}_i^{(k)}, \widehat{\boldsymbol{f}}_i^{(k)})]\right| = \left|\partial_{\mu_i}(t_k - t_{k-1})\lambda_i(t_k) - x_{i,k}\log\lambda_i(t_k) + \frac{\omega_i}{2}[\widehat{\mu}_i^{(k)}]^2\right|$$
$$= \rho_k + \omega_i\widehat{\mu}_i^{(k)} \leq |\delta - \mu_{\min}^{-1}| + \omega_i\widehat{\mu}_i^{(k)}, \tag{23}$$

where the last step uses the result $\rho_k \leq |\delta - \mu_{\min}^{-1}|$, which is a direct consequence from Assumption D.1. By the update rule of $\widehat{\mu}_i^{(k)}$, we can see that if $\widehat{\mu}_i^{(k)} \leq \omega_i^{-1}|\delta - \mu_{\min}^{-1}|$, then

$$\widehat{\mu}_i^{(k+1)} \leq \widehat{\mu}_i^{(k)}(1 - \omega_i\eta_k) + \eta_k|\delta - \mu_{\min}^{-1}| \leq (1 - \omega_i\eta_k)\omega_i^{-1}|\delta - \mu_{\min}^{-1}| + \eta_k|\delta - \mu_{\min}^{-1}|$$
$$= \omega_i^{-1}|\delta - \mu_{\min}^{-1}|.$$

Therefore by Assumption D.2 and mathematical induction, $\widehat{\mu}_i^{(k)} \leq \omega_i^{-1}|\delta - \mu_{\min}^{-1}|$ for every $k \geq 0$. Combining this result with (23), we have

$$\left|\partial_{\mu_i} l_{i,k}[\lambda_i^{(z)}(\widehat{\mu}_i^{(k)}, \widehat{\boldsymbol{f}}_i^{(k)})]\right| \leq 2|\delta - \mu_{\min}^{-1}|. \tag{24}$$

With (22), (24) and (21), we have

$$2\sum_{k=1}^{M(t)}\left(l_{i,k}[\lambda_i^{(z)}(\widehat{\mu}_i^{(k)}, \widehat{\boldsymbol{f}}_i^{(k)})] - l_{i,k}[\lambda_i^{(z)}(\mu_i, \widehat{\boldsymbol{f}}_i^{(k)})]\right)$$

$$\leq -\omega_i\sum_{k=1}^{M(t)}\left(\mu_i - \widehat{\mu}_i^{(k)}\right)^2 + \sum_{k=1}^{M(t)}\frac{1}{\eta_k}\left[\left(\widehat{\mu}_i^{(k)} - \mu_i\right)^2 - \left(\widehat{\mu}_i^{(k+1)} - \mu_i\right)^2\right] +$$

$$+ \sum_{k=1}^{M(t)}\eta_k\left[\partial_{\mu_i} l_{i,k}[\lambda_i^{(z)}(\widehat{\mu}_i^{(k)}, \widehat{\boldsymbol{f}}_i^{(k)})]\right]^2$$

$$= \sum_{k=1}^{M(t)}\left[\frac{1}{\eta_k} - \frac{1}{\eta_{k-1}} - \omega_i\right]\left(\mu_i - \widehat{\mu}_i^{(k)}\right)^2 + 4|\delta - \mu_{\min}^{-1}|^2\sum_{k=1}^{M(t)}\eta_k$$

$$\leq 4|\delta - \mu_{\min}^{-1}|^2\sum_{k=1}^{M(t)}\eta_k,$$

where in the last step, we have invoked the assumption that when $\omega_i \geq \zeta$, and $\eta_k = 1/(k\zeta + b)$ for $k > 0$ [3], $\eta_k^{-1} - \eta_{k-1}^{-1} \leq \omega_i$. Furthermore,

$$\sum_{k=1}^{M(t)}\eta_k \leq \zeta^{-1}(1 + \log M(t)).$$

Hence, plugging this result into the previous equation, we have

$$\sum_{k=1}^{M(t)}\left(l_{i,k}[\lambda_i^{(z)}(\widehat{\mu}_i^{(k)}, \widehat{\boldsymbol{f}}_i^{(k)})] - l_{i,k}[\lambda_i^{(z)}(\mu_i, \widehat{\boldsymbol{f}}_i^{(k)})]\right) \leq 2\zeta^{-1}|\delta - \mu_{\min}^{-1}|^2(1 + \log M(t)),$$

completing the proof of Step 1.

**Step 2: Proving equation** (19). The proof of (19) follows the same procedure as the proof of (18). However, proving the counterpart of (24) is more complicated. We stated it in Lemma 1, and we now formally prove it.

**Step 2.1: Proof of Lemma 1.**

Recall from equation (9) that, at the $k$-th update epoch, the update rule for $\widehat{f}_{i,j}^{(k)}$ can be written as

$$\widehat{f}_{i,j}^{(k+1)} = -\eta_k \left[ (t_k - t_{k-1}) - \frac{x_{i,k}}{\lambda_i^{(z)} \left( \widehat{\mu}_i^{(k)}, \widehat{\boldsymbol{f}}_i^{(k)} \right)} \right] \sum_{\tau_{j,n} \in [t_k - z, t_k)} K(t_k - \tau_{j,n}, \cdot)$$
$$+ (1 - \eta_k \zeta_{i,j}) \widehat{f}_{i,j}^{(k)},$$

where, by Assumption 2.2, $K(x, x) \leq 1$ for all $x \in \mathbb{R}$. Since we have used the truncated intensity function $\lambda_i^{(z)}$, we have, by triangle inequality,

$$\left\| \sum_{\tau_{j,n} \in [t_k - z, t_k)} K(t_k - \tau_{j,n}, \cdot) \right\|_{\mathcal{H}}^2 \leq \left[ \sum_{\tau_{j,n} \in [t_k - z, t_k)} \| K(t_k - \tau_{j,n}, \cdot) \|_{\mathcal{H}} \right]^2$$

$$= \left[ \sum_{\tau_{j,n} \in [t_k - z, < t_k)} K(t_k - \tau_{j,n}, t_k - \tau_{j,n}) \right]^2 \leq \kappa_z^2,$$

where $z$ is the window size that is selected at the beginning of the algorithm. Here, we have used the assumption that the number of arrivals within $[t_k - z, t_k)$ is upper bounded by $\kappa_z$, by Assumption 2.1. In addition, by the design of the algorithm, we always have

$$\lambda_i^{(z)} \left( \widehat{\mu}_i^{(k)}, \widehat{\boldsymbol{f}}_i^{(k)} \right) \geq \mu_{\min}.$$

Hence, when $x_{i,k} = 1$,

$$\left\| \widehat{f}_{i,j}^{(k+1)} \right\|_{\mathcal{H}} \leq (1 - \eta_k \zeta_{i,j}) \left\| \widehat{f}_{i,j}^{(k)} \right\|_{\mathcal{H}} + \left| \eta_k \left( t_k - t_{k-1} - \frac{x_{i,k}}{\lambda_i^{(z)} \left( \widehat{\mu}_i^{(k)}, \widehat{\boldsymbol{f}}_i^{(k)} \right)} \right) \right| \kappa_z$$

$$\leq (1 - \eta_k \zeta_{i,j}) \left\| \widehat{f}_{i,j}^{(k)} \right\|_{\mathcal{H}} + \eta_k \left| \delta - \mu_{\min}^{-1} \right| \kappa_z, \tag{25}$$

where in the last step of (25), we have used the technical assumption D.1. When the algorithm initializes with $\widehat{f}_{i,j}^{(0)}$ satisfies Assumption D.2, i.e.,

$$\left\| \widehat{f}_{i,j}^{(0)} \right\|_{\mathcal{H}} \leq \kappa_z \zeta_{i,j}^{-1} \left| \delta - \mu_{\min}^{-1} \right|,$$

we can use induction and (25) to show that every $\widehat{f}_{i,j}^{(k)}$ satisfies the above bound. In addition, by (9), we have

$$\left\| \partial_{f_{i,j}} l_{i,k} \left( \widehat{f}_{i,j}^{(k)} \right) \right\|_{\mathcal{H}} \leq 2\kappa_z \left| \delta - \mu_{\min}^{-1} \right|.$$

Similarly, when $x_{i,k} = 0$, the term $\mu_{\min}^{-1}$ vanishes because $x_{i,k} = 0$, and hence we reach the desired statement.

**Step 2.2: Strong convexity of the loss function.** The instantaneous loss function is strongly convex in the following sense:

$$l_{i,k}[\lambda_i^{(z)}(\mu_i, \boldsymbol{f}_i)] \geq l_{i,k}[\lambda_i^{(z)}(\mu_i, \widehat{\boldsymbol{f}}_i^{(k)})] + \sum_{j=1}^{p} \left\langle \partial_{f_{i,j}} l_{i,k}(\lambda_i^{(z)}(\mu_i, \widehat{\boldsymbol{f}}_i^{(k)})), f_{i,j} - \widehat{f}_{i,j}^{(k)} \right\rangle +$$

$$+ \sum_{j=1}^{p} \frac{\zeta_{i,j}}{2} \left\| f_{i,j} - \widehat{f}_{i,j}^{(k)} \right\|_{\mathcal{H}}^2.$$

In particular, the instantaneous loss function is strongly convex with respect to any one of the $f_{i,j}(t)$s and $\|\cdot\|_{\mathcal{H}}^2$ when the remaining $p-1$ are fixed. The proof follows directly from the strong convexity of $\|\cdot\|_{\mathcal{H}}^2$.

**Step 2.3: Proof of** (19). We now prove (19). By the strong convexity of the instantaneous loss function proved in Step 2.2, we have

$$2l_{i,k}[\lambda_i^{(z)}(\mu_i, \boldsymbol{f}_i)] \geq 2l_{i,k}[\lambda_i^{(z)}(\mu_i, \widehat{\boldsymbol{f}}_i^{(k)})] + 2\sum_{j=1}^p \left\langle \partial_{f_{i,j}} l_{i,k}(\widehat{f}_{i,j}^{(k)}), f_{i,j} - \widehat{f}_{i,j}^{(k)} \right\rangle_{\mathcal{H}} + \tag{26}$$

$$+ \sum_{j=1}^p \zeta_{i,j} \left\| f_{i,j} - \widehat{f}_{i,j}^{(k)} \right\|_{\mathcal{H}}^2. \tag{27}$$

This can be written as follows

$$2l_{i,k}[\lambda_i^{(z)}(\mu_i, \widehat{\boldsymbol{f}}_i^{(k)})] - 2l_{i,k}[\lambda_i^{(z)}(\mu_i, \boldsymbol{f}_i)] \leq 2\sum_{j=1}^p \left\langle \partial_{f_{i,j}} l_{i,k}(\widehat{f}_{i,j}^{(k)}), \widehat{f}_{i,j}^{(k)} - f_{i,j} \right\rangle -$$

$$- \sum_{j=1}^p \zeta_{i,j} \left\| f_{i,j} - \widehat{f}_{i,j}^{(k)} \right\|_{\mathcal{H}}^2. \tag{28}$$

For any $j \in \{1, \ldots, p\}$, since $\widehat{f}_{i,j}^{(k+1)} = \Pi[\widehat{f}_{i,j}^{(k)} - \eta_k \partial_{f_{i,j}} l_{i,k}[\lambda_i^{(z)}(\mu_i, \widehat{\boldsymbol{f}}_i^{(k)})]]$ and $\Pi[\cdot]$ is contractive, we have

$$\left\| \widehat{f}_{i,j}^{(k+1)} - f_{i,j} \right\|_{\mathcal{H}}^2 \leq \left\| \widehat{f}_{i,j}^{(k)} - f_{i,j} - \eta_k \partial_{f_{i,j}} l_{i,k}[\lambda_i^{(z)}(\mu_i, \widehat{\boldsymbol{f}}_i^{(k)})] \right\|_{\mathcal{H}}^2$$

$$= \left\| \widehat{f}_{i,j}^{(k)} - f_{i,j} \right\|_{\mathcal{H}}^2 + \eta_k^2 \left\| \partial_{f_{i,j}} l_{i,k}[\lambda_i^{(z)}(\mu_i, \widehat{\boldsymbol{f}}_i^{(k)})] \right\|_{\mathcal{H}}^2 -$$

$$- 2\eta_k \left\langle \partial_{f_{i,j}} l_{i,k}[\lambda_i^{(z)}(\mu_i, \widehat{\boldsymbol{f}}_i^{(k)})], \widehat{f}_{i,j}^{(k)} - f_{i,j} \right\rangle_{\mathcal{H}}.$$

Therefore,

$$2\left\langle \partial_{f_{i,j}} l_{i,k}[\lambda_i^{(z)}(\mu_i, \widehat{\boldsymbol{f}}_i^{(k)})], \widehat{f}_{i,j}^{(k)} - f_{i,j} \right\rangle_{\mathcal{H}} \leq \frac{1}{\eta_k} \left[ \left\| \widehat{f}_{i,j}^{(k)} - f_{i,j} \right\|_{\mathcal{H}}^2 - \left\| \widehat{f}_{i,j}^{(k+1)} - f_{i,j} \right\|_{\mathcal{H}}^2 \right] +$$

$$+ \eta_k \left\| \partial_{f_{i,j}} l_{i,k}[\lambda_i^{(z)}(\mu_i, \widehat{\boldsymbol{f}}_i^{(k)})] \right\|_{\mathcal{H}}^2. \tag{29}$$

Using Lemma 1, we have

$$\eta_k \left\| \partial_{f_{i,j}} l_{i,k}[\lambda_i^{(z)}(\mu_i, \widehat{\boldsymbol{f}}_i^{(k)})] \right\|_{\mathcal{H}}^2 \leq 4\eta_k \kappa_z^2 |\delta - \mu_{\min}^{-1}|^2$$

when $x_{i,k} = 1$, and

$$\eta_k \left\| \partial_{f_{i,j}} l_{i,k}[\lambda_i^{(z)}(\mu_i, \widehat{\boldsymbol{f}}_i^{(k)})] \right\|_{\mathcal{H}}^2 \leq 4\eta_k \kappa_z^2 \delta^2 \leq 4\eta_k \kappa_z^2 |\delta - \mu_{\min}^{-1}|^2$$

when $x_{i,k} = 0$.

We now proceed to final step, which sums (28) over $k \in \{1, \ldots, M(t)\}$ and then combines the result with (29) summed over $j \in \{1, \ldots, p\}$. To obtain stronger intuition, we choose to use the uniform upper bound $4\eta_k \kappa_z^2 |\delta - \mu_{\min}^{-1}|^2$ for $\eta_k \|\partial_{f_{i,j}}\| l_{i,k}[\lambda_i^{(z)}(\mu_i, \widehat{\boldsymbol{f}}_i^{(k)})]$, which holds for all values of $x_{i,k}$. We thus obtain

$$2\sum_{k=1}^{M(t)} \left( l_{i,k}[\lambda_i^{(z)}(\mu_i, \widehat{\boldsymbol{f}}_i^{(k)})] - l_{i,k}[\lambda_i^{(z)}(\mu_i, \boldsymbol{f}_i)] \right) \leq \sum_{k=1}^{M(t)} \sum_{j=1}^p \left\| \widehat{f}_{i,j}^{(k)} - f_{i,j} \right\|_{\mathcal{H}}^2 \left( \frac{1}{\eta_k} - \frac{1}{\eta_{k-1}} - \zeta_{i,j} \right) +$$

$$+ 4\kappa_z^2 |\delta - \mu_{\min}|^2 \sum_{k=1}^{M(t)} \sum_{j=1}^p \eta_k. \tag{30}$$

Since $\eta_k = 1/(k\zeta + b)$, we obtain

$$\sum_{k=1}^{M(t)} \eta_k \leq \zeta^{-1}(1 + \log M(t)).$$

Furthermore, $1/\eta_k - 1/\eta_{k-1} - \zeta_{i,j} \leq 0$. Therefore, substituting the above inequalities into (30), we get

$$\sum_{k=1}^{M(t)} \left( l_{i,k}[\lambda_i^{(z)}(\mu_i, \widehat{\boldsymbol{f}}_i^{(k)})] - l_{i,k}[\lambda_i^{(z)}(\mu_i, \boldsymbol{f}_i)] \right) \leq 2p\zeta^{-1}\kappa_z^2|\delta - \mu_{\min}^{-1}|^2(1 + \log M(t)).$$

**Step 3.** The overall regret bound can be obtained by adding (18) and (19).

### D.1 Proof of Corollary 1

From the result of Proposition 1, we have

$$\left| L_{i,t}^{(\delta)}(\lambda_i^{(z)}) - L_{i,t}(\lambda_i) \right| \leq (1 + \frac{\kappa_1}{\mu_{\min}})N(t - z)\varepsilon(z) + \delta N(t)\varepsilon'(0).$$

Using the above inequality, the results of Theorem 1, and the triangle inequality, we obtain

$$\sum_{k=1}^{M(t)} \left( l_{i,k}(\lambda_i^{(z)}(\widehat{\mu}_i, \widehat{\boldsymbol{f}}_i^{(k)})) - l_{i,k}(\lambda_i(\mu_i, \boldsymbol{f}_i)) \right) \leq (C_1 + C_2)(1 + \log M(t)) + C_3 N(t),$$

where $C_1 = (1 + \zeta)^{-2}|\delta - \mu_{\min}^{-1}|^2 + 2\kappa_z^2\delta^2 p$ and $C_2 = 2\kappa_z^2\mu_{\min}^{-2} - 4\kappa_z^2\delta\mu_{\min}^{-1}$.

## E  Projection Procedure and Proof of Proposition 2

In this section, we derive the detailed procedure of using polynomial kernels comparing to any other kernels in general. Recall that when using any kernel in general, the projection operation is a QP problem, and in order to ensure $\widehat{f}_{i,j}^{(k)}(t) \geq 0$ for any $t \in [0, z]$, the problem is subject to infinite number of constraints: for every $t > 0$, we must have $\widehat{f}_{i,j}^{(k)}(t) \geq 0$, which makes the exact projection onto the subspace of positive functions intractable.

Surprisingly, however, when using 1-dimensional polynomial kernels (e.g., $K(x, y) = (1 + xy)^4$), one can convert the projection problem to an SDP using the following procedure. First, notice that the original projection problem can be written as

$$\widehat{f}_{i,j}^{(k+1)} = \underset{f \in \mathcal{H}, f(t) \geq 0}{\operatorname{argmin}} ||\widehat{f}_{i,j}^{(k+\frac{1}{2})} - f||_{\mathcal{H}}^2 = \underset{f \in \mathcal{H}, f(t) \geq 0}{\operatorname{argmin}} -2\langle f, \widehat{f}_{i,j}^{(k+\frac{1}{2})}\rangle_{\mathcal{H}} + ||f||_{\mathcal{H}}^2. \qquad (31)$$

By the representer theorem, we also have $\widehat{f}_{i,j}^{(k+\frac{1}{2})}(\cdot) = \sum_{s \in \mathcal{S}} a_s K(s, \cdot)$, where

$$\mathcal{S} = \cup_{r \leq k}\{t_r - \tau_{j,n} : t_r - z \leq \tau_{j,n} < t_r\}.$$

Plugging the expression of $\widehat{f}_{i,j}^{(k+\frac{1}{2})}$ into (31), and using the fact that $\langle K(s, \cdot), f\rangle_{\mathcal{H}} = f(s)$, we have

$$\underset{f \in \mathcal{H}, f(t) \geq 0}{\operatorname{argmin}} -2\sum_{s \in \mathcal{S}} a_s f(s) + ||f||_{\mathcal{H}}^2. \qquad (32)$$

Let $\mathcal{H}$ be the RKHS with kernel $K(x, y) = (1 + xy)^{2d}$, for some integer $d$. By the solution of Hilbert's 17th problem [7], we know that a 1-dimensional and $2d$-degree polynomial is non-negative if and only if it can be written as the sum of squares of $d$-degree polynomials, i.e., a quadratic form of $d$-degree polynomials. This allows us to substitute the constraint in (32) with the following:

$$f \in \{\phi^\top(x)\mathbf{Q}\phi(x); \ \mathbf{Q} \succeq 0\} \subset \mathcal{H},$$

where $\phi(x)$ is the feature map of the kernel function $K'(x,y) = (1 + xy)^d$, which satisfies $\phi^\top(x)\phi(y) = K'(x,y)$.

Finally, by the representer Theorem in [5] for positive functions, we obtain that the minimizer to (32) is of the form $\sum_{s\in\mathcal{S}} b_s K(s,\cdot)$. Hence, (32) can be written as

$$\underset{\mathbf{b}\in\mathbb{R}^{|\mathcal{S}|}}{\operatorname{argmin}} \ -2\mathbf{a}^\top\mathbf{K}\mathbf{b} + \mathbf{b}^\top\mathbf{K}\mathbf{b} \tag{33}$$

$$\text{s.t.} \ \ f(x) = \sum_{s\in\mathcal{S}} b_s K(s,x) = \phi^\top(x)\mathbf{Q}\phi(x), \ \text{for some } \mathbf{Q} \succeq 0.$$

Upon simple manipulations, the above problem further reduced to

$$\underset{\mathbf{b}\in\mathbb{R}^{|\mathcal{S}|}}{\operatorname{argmin}} \ -2\mathbf{a}^\top\mathbf{K}\mathbf{b} + \mathbf{b}^\top\mathbf{K}\mathbf{b} \tag{34}$$

$$\text{s.t.} \ \ \mathbf{G} \cdot \operatorname{diag}(\mathbf{b}) + \operatorname{diag}(\mathbf{b}) \cdot \mathbf{G} \succeq 0,$$

where $\mathbf{K} = [K(s,s')]$, and $\mathbf{G} = [\phi^\top(s)\phi(s')] = [K'(s,s')]$ are Gramian matrices with $s$ and $s'$ coming from $\mathcal{S}$.

# F  Experiment Details

In this section, we present more details on the experiment, including the settings of parameters, the complete set of estimates for the synthetic data, in Figure 4, and the complete set of estimates for real data, in Figure 5.

**Experiment settings.** For the synthetic data, we generated two sets of data. The first set of data has a larger time horizon, $T = 10^5$, and the simulation results are used to examine the goodness-of-fit, and run time comparison. We refer to this set of data as "dataset 1".

To evaluate the effects of the hyperparameters, we also generated a second set of data, which contains 100 trials of independently generated realizations of the MHP, with a smaller time horizon $T = 10^4$. These data, which we refer to as "dataset 2", contribute to the simulation results presented in Table 1.

**Parameters used on "dataset 1".** For NPOLE-MHP, we set the discretization level $\delta = 0.05$, the window size $z = 3$, the step size $\eta_k = (k\delta/20 + 100)^{-1}$, the regularization coefficient $\zeta_{i,j} = 10^{-8}$, and we use a Gaussian kernel whose bandwidth is 0.2. To speed up the algorithm, we use $f_{i,j}(t) = g_{i,j}^2(t)$ setting to circumvent the projection step, and we round $t_k - \tau_{j,n}$ onto a discretized set of points of the form $0.02k$ for $k \in \{1, \ldots, \lfloor 50z \rfloor\}$. This allows us to compute the Gramian matrix of the kernel beforehand. We simulated the benchmark algorithms on fine-tuned parameters. For DMD and OGD algorithms in [15], we set $f_{i,j}(t) = \alpha_{i,j}\exp\{-2t\}$, set $\mathbf{W}$ as the coefficient matrix, $\delta = 0.05$, $\eta = 0.01(10^5/\delta)^{-1/2}$ as the step size for all the gradient descent involved. For the E-M algorithm of [27], we use the truncated version of the intensity function, and set the number of kernels as well as the kernel bandwidth to be the same as NPOLE-MHP. This potentially improves the performance of [27] since all the data can now be used for learning. We set the coefficient for the $\ell_1$ regularization to be 100, and the coefficient for $\ell_{1,2}$ regularization to be 10. The pairwise similarity is not considered. Finally, we set 8 rounds of iterations for both inner and outer loops of the algorithm.[4] Since the DMD algorithm requires knowing the true underlying $\boldsymbol{\mu}$, we assumed that $\boldsymbol{\mu}$ was available to all algorithms for fair comparison.

**Parameters used on "dataset 2".** For each choice of $\delta$ and $\zeta$, we choose $\eta = 1/(k/2000 + 10)$, while remaining the initialization, bandwidths and landmarks of the kernels, as well as the window size to be the same as previously. The change in the step size takes into consideration that the number of events is lesser than in "dataset 1", implying the need for a potentially larger step size in the beginning. The step size is selected by grid search over 1 trial of data by minimizing the cumulative loss under $\delta = 0.01$ and $\zeta = 10^{-8}$. For the MLE and MLE-SGLP, we use 5 rounds of inner iterations

Figure 4: Performance of different algorithms for estimating $\boldsymbol{F}$. The subplot at the $i$-th row and $j$-th column shows the performance on $f_{i,j}(t)$. For each subplot, the horizontal axis ranges from 0 to $z = 3$, while the vertical axis covers $[0, 1]$.

and 5 rounds of outer iterations to control the amount of time for simulation. As a consequence, the superior results of NPOLE-MHP, demonstrated in Table 1 in terms of the "average $L_1$ error", should not be interpreted as the superiority of NPOLE-MHP over the maximum likelihood algorithm. The worse result produced by the MLE algorithm is a result of limited computational resource.

**Heat maps for real data.** For the real data, we compare the values of $\|\widehat{f}_{i,j}\|_{L_1[0,z]}$ by converting them into color maps (Figure 5). Top left corner, $\|\widehat{f}_{i,j}\|_{L_1[0,z]}$ is computed using the output of MLE of [27] with 8 outer loops and 8 inner loops, respectively, using 18 days of the meme-tracking dataset. For the rest of the three plots, we calculate $\|\widehat{f}_{i,j}\|_{L_1[0,z]}$ using the output of NPOLE-MHP with different step sizes. It can be seen that NPOLE-MHP generates similar sparsity patterns to that of MLE where the diagonal dominates.

MLE estimate with 8 outer loops and 8 inner loops.

NPOLE-MHP esitmate with $\eta_k = 1/(k\zeta + 400)$.

NPOLE-MHP esitmate with $\eta_k = 1/(k\zeta + 600)$.

NPOLE-MHP esitmate with $\eta_k = 1/(k\zeta + 800)$.

Figure 5: NPOLE-MHP and MLE: a color map comparison.

## Footnotes

[3] we assume $1/\eta_0 = 0$ since $\widehat{\mu}_i^{(0)}$ was not involved in the summation.

[4]The algorithm of [27] is implemented based on a slight modification of the code provided by the authors.