[Reviews · NeurIPS 2017]

Reviewer 1



This paper describes an algorithm for optizimization of Hawkes process parameters in on-line settings, where non-parametric form of a kernel is learnt. The paper reports a gradient approach to optimization, with theoretical analysis thereof. In particular, the authors provide: a regret bound, justification for simplification steps (discretization of time and truncation of time over which previous posts influence a new post), an approach to a tractable projection of the solution (a step in the algorithm), time complexity analysis. The paper is very well written, which is very helpful given it is mathematically involved. I found it tackling an important problem (on-line learning is important for large scale datasets, and non-parametricity is a very reasonable setting when it is hard to specify a reasonable kernel form a priori). The authors provide theoretical grounding for their approach. I enjoyed this paper very much and I think it is going to be influencial for practical application of Hawkes processes. Some comments: - Can you comment on the influence of hyperparameter values (regularization coefficients, potentially also discretization step)? In experiments you set them a-priori to some fixed values, and it is not uncommon in the experiments that values of regularization hyperparameters have a huge influence on the experiments. - Your experiments are quite limited, especially the synthetic experiments where many values are fixed and a single matrix of influence forms is considered. I think it would be much more compelling if some simulation was considered, where values would be repeatedly drawn. Also, why do you think this specific setting is interesting/realistic? - I suggest making the font in Figures larger (especially for Figure 2 it is too small), it doesn't look good now. This is especially annoying in the supplementary (Figure 4).

Reviewer 2



The paper deals with an interesting and important problem -- how to learn the triggering functions of multivariate Hawkes processes in an online fashion. The proposed method gets rid of the parametric assumptions of the triggering functions and assumes that the triggering functions come from the reproducing kernel Hilbert space and is a nonparametric method. It is proven that the proposed algorithm can achieve a sublinear regret bound. And the proposed nonparametric method is numerically evaluated on synthetic and real datasets. Comments: 1. The paper is overall well written before section 3. The current algorithm section and theoretical analysis section are not easy to follow. It seems that the authors have skipped some necessary details which makes it a bit difficult for readers to fully understand their algorithm and their theoretical analyses. 2. The notations in section 3 are a bit heavy. There are quite a few tuning parameters in the proposed algorithm. From the current descriptions it is not clear how to set these parameters in practice. For example, how to determine the regularization coefficients and how to choose the step size in your algorithm? Will the values of these parameter influence the final output? It is not clear in section 3. In proposition 2, only polynomial kernels are considered. How about other types of reproducing kernel? Need more discussions here. 3. For the theoretical section, is your current loss function convex? I think this is an important issue and more clarifications are needed. 4. The numerical evaluation of the algorithm can be improved. For example, a bigger size of problem can be considered. 5. This submission is using a template for NIPS 2016.

Reviewer 3



The authors propose an online algorithm for inferring the triggering functions among different dimensions of an MHP. Due to the vast application of Hawkes processes in modeling real-world problems with streaming data and the importance of revealing the interrelation among different dimensions in these scenarios, the problem of the paper is interesting and important. However, the authors didn’t motivate the problem well and their experiments are not satisfying enough. I think the paper took a principled approach to estimate the log-likelihood of MHPs as a function of only triggering kernels by discretizing the time axis and using an online kernel learning method. It has a strong theoretical basis. However, I have the following concerns: 1. Although the authors provide good theoretical results about the efficiency of the proposed algorithm, there are no experiments showing the scalability of the model in real-world scenarios. 2. The experimental results need further improvements. For example, there should be a theoretical or experimental analysis of the sensitivity of the proposed algorithm performance with respect to f^(0), \mu^(0), step size and other parameters. 3. The performance of MLE-SGLP and MLE seems more or less the same as the proposed method. A quantitative measure would clarify the differences. 4. The analysis of the performance of the algorithm with respect to the number of training data is missing. In summary, the paper has a strong theoretical basis, but lacking in some theoretical/experiential analysis that mentioned above, which doubt the superiority of the proposed algorithm with respect to the other methods like MLE-SGLP and MLE.